# Design and Evaluation of Sensor Housing for Boundary Layer Profiling Using Multirotors

**DOI:** 10.3390/s19112481

**Published:** 2019-05-30

**Authors:** Ashraful Islam, Adam L. Houston, Ajay Shankar, Carrick Detweiler

**Affiliations:** 1Department of Mechanical & Materials Engineering, University of Nebraska-Lincoln, Lincoln, NE 68588, USA; 2Department of Computer Science & Engineering, University of Nebraska-Lincoln, Lincoln, NE 68588, USA; ashankar@cse.unl.edu (A.S.); carrick@cse.unl.edu (C.D.); 3Department of Earth & Atmospheric Sciences, University of Nebraska-Lincoln, Lincoln, NE 68588, USA; ahouston2@unl.edu

**Keywords:** sensor housing, multirotor Unmanned Aerial System, TH sensor housing, boundary layer profile

## Abstract

Traditional configurations for mounting Temperature–Humidity (TH) sensors on multirotor Unmanned Aerial Systems (UASs) often suffer from insufficient radiation shielding, exposure to mixed and turbulent air from propellers, and inconsistent aspiration while situated in the wake of the UAS. Descent profiles using traditional methods are unreliable (when compared to an ascent profile) due to the turbulent mixing of air by the UAS while descending into that flow field. Consequently, atmospheric boundary layer profiles that rely on such configurations are bias-prone and unreliable in certain flight patterns (such as descent). This article describes and evaluates a novel sensor housing designed to shield airborne sensors from artificial heat sources and artificial wet-bulbing while pulling air from *outside* the rotor wash influence. The housing is mounted *above* the propellers to exploit the rotor-induced pressure deficits that passively induce a high-speed laminar airflow to aspirate the sensor consistently. Our design is modular, accommodates a variety of other sensors, and would be compatible with a wide range of commercially available multirotors. Extensive flight tests conducted at altitudes up to 500 m Above Ground Level (AGL) show that the housing facilitates reliable measurements of the boundary layer phenomena and is invariant in orientation to the ambient wind, even at high vertical/horizontal speeds (up to 5 m/s) for the UAS. A low standard deviation of errors shows a good agreement between the ascent and descent profiles and proves our unique design is reliable for various UAS missions.

## 1. Introduction

High-resolution profiles of the Atmospheric Boundary Layer (ABL) are critical for obtaining measurements important for a range of microscale and mesoscale phenomena. These profile measurements are conventionally performed using ground-based balloon launches [1,2] and parachute-based dropsondes [3]. Recent technological advancements have enabled the use of Unmanned Aerial Systems (UASs) to perform targeted, controlled, and frequent profiles using fixed wings [4] and multirotors [5,6]. Both fixed-wing and multirotor UASs offer distinct advantages over passive balloon and parachute systems in terms of the ease, frequency, and spatiotemporal resolution of observations. More specifically, multirotors have flight profiles that allow easy operation, a geometry that offers convenient placement for a variety of sensors, and a payload capacity that even permits heavier sensors (ultrasonic anemometer, particle counter, aerosol sensor, etc.) to be airborne. Furthermore, their relatively small launch/land footprint makes them suitable for use in remote locations without the need for runways [7]. As a result, their use for a variety of profiling applications such as the measurement of Temperature–Humidity (TH) data [8,9], wind-estimation [10,11,12,13], air quality measurements [14,15], etc. has increased significantly in recent years.

This paper develops the housing for TH measurement, which is shown mounted and fully assembled on a multirotor in Figure 1. This housing outperforms traditional mounting configurations for TH sensors (underneath the body, under the propeller-arm, etc. described in Section 2.2, and Table 1) in terms of data consistency, correctness, and reliability. The performance improvements of the housing comes from mitigating three primary sources of measurement errors of a TH sensor mounted on a multirotor (error/uncertainty due to sensor response time, radiation error, and undesirable mixing) by pulling air through the housing sourced away from a UAS-disturbed air flow and by maintaining a high airflow speed. Furthermore, the modular and light-weight design makes it convenient to set up in the field and adaptable to a range of UASs and sensors. The housing system imposes minimal constraints on regular flight operations.

Obtaining a precise air temperature is particularly challenging as uncertainties can arise from multiple sources such as the diameter of the sensor, the airflow speed for aspiration, the sensor self-heating, the sensor response time, etc. [16,17]. Additionally, undesired mixing in the air, such as that of propeller flow field, can cause a misrepresentation of the observation level or an undesired smoothing of atmospheric features. As a result, the quality of the measurements made by the sensor is also extremely dependent on its placement on the UAS’s body [18,19] and aspiration speed [20]. TH sensors are particularly sensitive to a lack of aspiration and inadequate shielding from insolation and precipitation, the latter of which can cause erroneous wet-bulbing. A TH sensor mounted underneath the rotor of the multirotor, for example, can introduce bias into its measurements due to insolation, compressional, and frictional heating [18,21]. The sensor will also have inconsistent aspiration due to rotor turbulence [22,23]. Moreover, because the air beneath the rotors originates from the area above the aircraft, observations using traditional mounting have uncertainty in the actual observation level. These biases are particularly acute in descent profiles, making it extremely difficult to ensure the minimum vertical resolution to accurately capture atmospheric phenomena [24].

Consequently, several TH-measurement approaches rely only on the ascent data [24,25,26] while using only minimal measures to shield the sensor. Ideally, placing the sensor farther away from the body of the UAS increases the validity of the recorded observations [15]. Although many different probes for air/gas parameter measurement exist where flow is sucked inside a tube [27], our method does not rely on vacuum pump/atmospheric wind to induce the flow. We use an expander-reducer strategy to exploit the low pressure region on top of the propeller to induce a high speed flow that satisfies the requirements of TH sensors. However, doing so requires a careful design of a sensor housing that fulfills all the requirements of sensors and UAS and still maintains reasonable margins of safety for UAS flight operations.

In this paper, we present a novel housing design for mounting TH sensors that not only ensures environmental shielding but also provides consistent aspiration throughout the mission despite the changes in airflow field around the UAS. Our design innovation ensures that measurements are taken from the air far away from the body of the aircraft and near the actual observation level. The housing passively uses the suction induced by the propellers to create a high-speed laminar flow inside the sensor housing tube.

We present our results from various field deployments conducted over four days during the Lower Atmospheric Process Studies at Elevation—a Remotely-piloted Aircraft Team Experiment (LAPSE-RATE) flight campaign as part of the 2018 annual meeting of the International Society for Atmospheric Research using Remotely-piloted Aircraft (ISARRA) [28].

Our evaluations focus on three fundamental contributions of the design: (1) reliable temperature–humidity sensor readings during both ascent *and* descent in vertical profiling missions at varying speeds; (2) invariance of the orientation of the housed sensors to ambient wind (or lack thereof); and (3) the ability to accurately capture atmospheric phenomena (such as an inversion layer) when compared to a radiosonde. We note that, although wind measurement is one of the parameters of interest in ABL studies, our sensor housing is not designed for atmospheric wind measurement. Our primary research goal is acquiring accurate temperature-humidity measurements using the proposed sensor housing. However, based on the findings of the paper, we have noted the importance of on-board wind measurements in the discussion of future work.

The rest of this paper is organized as follows: the Materials and Methods are presented in Section 2. In Section 3, the Results are presented, followed by a Discussion of the results in Section 4 and the Conclusion in Section 5.

## 2. Materials and Methods

This section presents an explanation of the design requirements of the sensor housing (Section 2.1), an evaluation of the sensor placement configurations (Section 2.2), a description of the system with detailed features (Section 2.3), a description of the sensor housing (Section 2.4), and a conclusion of the experimental methods (Section 2.5) to validate the effectiveness of the sensor housing.

### 2.1. Design Requirements

The design of a sensor housing system has several fundamental requirements that stem from the properties of the sensor and the flight characteristics of the UAS.

**Requirements of the Sensor:** TH sensors require sufficient aspiration, protection from external heat sources (such as insolation and UAS waste heat), and prevention of artificial wet-bulbing.
*Aspirating Sensors:* Naturally aspirated sensors are prone to radiative heating error [29]. The aspiration airspeed also affects the effective sensor response times [20]. Insufficient and inconsistent aspiration speeds introduce uncertainty in the measurement that could be impossible to deal with during quality checks/correction of the data. Sensor manufacturers often specify the sensor response time in the presence of a constant airflow. For instance, a nominal 5 m/s airflow is used for the response time of 1 s for iMet sensors (Table 2). However, when mounted on a system, the response time of the *system* could change based on the aspiration method and should be determined *as flown* [30].*Shielding from Heat Sources:* The sensors need to be protected from solar radiation and other heat sources such as the waste heat produced by the battery and motors and absorbed heat from solar radiation by the body of the UAS. Failure to do so will result in biased measurements, which can worsen the longer a UAS-mounted sensor is airborne. Having a sufficiently aspirated housing and proper placement of the sensor will reduce/eliminate the effect of these biases.*Preventing Artificial Wet-Bulbing:* The sensor housing needs to shield the sensor from precipitation and should not allow water to accumulate on the sensors. The accumulation of liquid water on the sensor can produce incorrect measurements of the temperature by the wet-bulbing effect. If the humidity sensor package is dependent on the temperature sensor for corrections, this will also skew the humidity measurements by an undesired saturation.

**Requirements of the UAS:** The general requirements from the perspective of the UAS are that the weight and size of the housing must be within the bounds of the UAS’s payload capacity. At the same time, the placement of the housing must not hinder safe and normal operations of the UAS.
*Payload Capacity:* The sensor housing and its mounting fixtures must not exceed the rated payload capacity of the aircraft. A higher payload weight also reduces the total flight time, thereby limiting the type of mission the UAS will be capable of performing.*Flight Dynamics:* The housing must not prevent the UAS from doing its normal maneuvers and must remain in safe operating conditions. Adding the sensor housing must not severely disrupt the balance of the UAS.

### 2.2. Evaluation of Sensor Placement Configurations on the UAS

The different sensor configurations considered in the paper are shown in Figure 2 and summarized in Table 1. One approach to placing the sensor would be to mount it directly underneath the propeller (configuration 1) to meet the aspiration requirements, but the flow field of a propeller is turbulent [22,23] and the resulting aspiration is unsteady. Sensors in this configuration can show bias due to frictional and compression heating by propellers and motor waste heat [18,21]. Also, airflow passing across the sensor originates from a large area above the UAS [22], making the location of actual measurements uncertain. Alternatively, the sensor can be placed on top of a propeller (configuration 4) where the air is relatively unmixed and the flow is comparatively laminar. However, as the UAS descends into turbulent and mixed air created by its propeller, this sensor configuration will be in the wake of the UAS, will be measuring data from an uncertain/mixed source, and will potentially experience uncertain aspiration if obstructed by the UAS body. Moreover, the resulting aspiration might be insufficient and unreliable as the airflow speed is dependent on the propeller dimension and relative distance from the motor.

Insufficient aspiration is also a potential issue if the UAS is to be used in a similar configuration as a radiosonde-weather-balloon where the sensor is passively mounted above or hanging from the UAS (configurations 2 and 3), but the airflow speed decreases drastically towards the center of the UAS [15]. Any sensor mounted outside of the housing (configurations 1–4) due to the lack of shielding also runs the risk of bias from insolation or erroneous wet-bulbing. The sensor mounted inside a horizontal cylinder in configurations 2 and 4 could provide radiation shielding and, in some cases, provide sufficient aspiration (only if the atmospheric wind speed/horizontal velocity of the UAS is high and in a specific orientation), but this will limit the use case of the multirotor in ABL measurements. Both of the configurations will still be in the wake of the UAS during descent, hence adding uncertainty to the observations during descent.

Sensors inside the housing, as shown in Figure 2 (corresponding to configurations 5–7), will be shielded from radiation and would maintain sufficient aspiration. Configuration 5 will have turbulent air and potential aspiration issues during descent. Configuration 6 (even though it is better than configuration 5) could be subjected to waste heat of the UAS body, and its air source will be in the wake of UAS during descent. Considering an extensive evaluation of configurations 1–6 in our previous effort [19,31], along with the fact that a sensor placed in configuration 7 will sample the air that is significantly less interfered by the UAS, configuration 7 would be the best choice.

Aspirating air flow through the housing could be induced using active or passive air draw. The active draw of air will require additional hardware like a fan, a battery, and associated electronics [21]. This also does not utilize what is already available in a UAS, i.e., the wind field from the propellers. Additional hardware will also make the housing heavier and will reduce the flight time of the UAS. Using the propeller of the UAS to draw air passively can facilitate a similar (or better) performance with less payload.

Since for the housing in configuration 7 the air source is far from the center of a multirotor, the extension has the potential to compromise the stability of the UAS by introducing an extra moment of inertia on one arm. To balance the moment of inertia, we choose to mount a second housing in a symmetric configuration on the opposite arm. We note that this may make the UAS less responsive to control inputs in the roll/pitch axis.

Based on the above design considerations, this paper uses symmetric configuration 7 for sensor housing (with passive air draw), as seen in Figure 1. As for a comparison against sensors without sensor housing, we choose configuration 3.

### 2.3. System Description

The primary components of the housing system, as shown in Figure 1, are the UAS, Data Acquisition System (DAQ), sensors, sensor housing, and support structure for sensor housing.

**UAS:** The UAS configuration is a “DJI Matrice 600 Pro” hexacopter platform equipped with a “DJI A3 Pro” flight control system. The dimensions of the system are 1668 mm × 1518 mm × 727 mm with propellers, frame arms, and a GPS mount unfolded (including landing gear). The maximum takeoff weight of the UAS is 15.5 kg (the maximum recommended payload is 5.5 kg) with a flight endurance of 35–40 min on a single set of six “DJI TB48S” batteries with no load. The manufacturer specified positioning accuracy is vertically ± 0.5 m and horizontally ± 1.5 m [32]. The maximum ascent and descent speeds are 5 m/s and 3 m/s respectively.

**DAQ:** All the experimental data are obtained and recorded onboard the UAS by an Odroid XU4 [33], which is a compact single-board computer with eight processing cores and 2 GB of RAM. The Odroid runs a Robot Operating System (ROS) [34] node that communicates with the autopilot on the UAS to collect the position, velocity, altitude, attitude, and other control information. It also interfaces with the TH sensors to obtain temperature and humidity measurements. Different ROS nodes collect data independently of each other and record the time stamp of when the ROS node collected the data and when the ROS core received the data, which simplifies the synchronization between the different sources of data node. A wireless serial interface to the Odroid (over 2.4 Ghz Xbee radios) allows two-way communication from a ground station, which is also useful for debugging and periodic sanity checks. The raw data file, recorded by the ROS, is retrieved over an ethernet connection and post-processed in MATLAB.

**Sensors:** Detailed specifications of the TH sensors used in the paper are described in Table 2. The primary sensors used for the experiments are iMet XQ2 and iMet XQ1 from InterMet Systems (Grand Rapids, MI, USA). InterMet sensors feature a self-contained sensor package designed for UASs that include atmospheric pressure, temperature, and humidity sensors. These sensor units also have built-in GPSs and an internal data logger with a rechargeable battery.

Additionally, a custom two-node sensor, called nimbus-pth, designed and built in our lab, was mounted on one of the UAS platforms. The nimbus-pth sensor has a high precision thermistor configured in a Wheatstone bridge and is read by a 10-bit analog-to-digital converter. The humidity sensor is an “SHT31” sensor, while the pressure sensor is an “MS5803” sensor. For experiments involving measurements of airflow speed inside the housing, a hot wire thermo-anemometer “Extech SDL350” is used inside the housing. The anemometer has a specified range of 0.2–25 m/s with a resolution of 0.01 m/s and accuracy of 5% of the reading.

The first ground truth reference used for the experiments is the instrumented van of Colorado University, Boulder, CO named MURC (Mobile UAS Research Collaboratory) [35]. As part of the LAPSE-RATE’2018 flight campaign intercomparison of multirotor and fixed wing platforms, 13 institutions and organizations used the measurement from the MURC tower as the ground truth to fly 35 UASs in a similar flight pattern. The MURC has a 15.2 m extendable tower that is equipped with Gill MetPak Pro Pressure, Temperature, Humidity (PTH) sensors. The MetPak Pro sensor has a manufacturer specified accuracy of (±0.1 ${}^{\circ }$C) for temperature and (±0.8% of RH) for humidity. The filtering process of MURC sensors are proprietary by manufacturer and not available.

Another ground truth used in this paper is radiosonde balloon observation data provided by the National Oceanic and Atmospheric Administration (NOAA). The balloons were deployed by NOAA official close to the UAS launching site by following standard deployment guidelines. The sensor used in the radiosonde is Vaisala RS92 (Vaisala Corporation, Vantaa, Finland) with a very fast response time for both temperature (<0.4 s) and humidity(<0.5 s). The radiosonde data presented in the paper is corrected and filtered for sensor response lag and other factors by proprietary Vaisala algorithms.

**UAS Sensor Mounting Configuration and Payload:** The sensor mounting configuration of the sensors is referred to as a configuration number as described in Section 2.2 and Figure 2. The placement configuration of the sensors used in the experiments for each of the UAS is as follows:

*UAS platform M600P1*: one XQ2 (code name: P1XQ2) is mounted inside the left sensor housing (configuration 7), and one XQ1 (code name: P1XQ1) is mounted inside the right sensor housing (configuration 7). The alternative setup used in some experiments replaces XQ1 with nimbus-pth (code name: P1Nimbus2) inside the housing. An additional nimbus-pth (code name: P1Nimbus1) is also placed under the body of the UAS without housing (configuration 3).

*UAS platform M600P2*: one XQ2 (code name: P2XQ2) is mounted inside the left sensor housing (configuration 7), one nimbus-pth (code name: P2Nimbus2) is mounted inside the right sensor sensor housing (configuration 7), and an additional nimbus-pth (code name: P2Nimbus1) is placed under the body of the UAS without housing (configuration 3).

This form of sensor placement facilitates an evaluation between the sensor placed inside the housing (configuration 7), as shown in Figure 3a, versus under the body of the UAS without housing (configuration 3), as displayed in Figure 3b.

For the configuration used in the experiments, the UAS’s payload was ∼1.8kg (housing with support structure and sensor—2×∼
720gm, onboard computer—140gm, misc cables, screws etc.—approx. 200gm) with a flight endurance of 20–25 min, which makes it capable of vertical profiling missions up to a 1000 m AGL at 2 m/s.

### 2.4. Sensor Housing

The sensor housing exploits the pressure deficit created on the top of a spinning multirotor propeller to induce high speed airflows inside itself. The passive-indirect draw of air, by positioning the housing air intake *above* the propeller, creates a laminar flow. The sensor housing is designed in accordance with a fluid dynamics textbook [36] to ensure that the housing maintains a smooth laminar flow throughout the housing with a low minor loss of flow. Besides maintaining a consistent aspiration for the sensor, the housing also shields it from incident solar radiation and other heat sources.

The sensor housing, as shown in Figure 4a,b, is made from seven components (with an identical inlet and outlet). The individual parts connect using mechanical screw-in threads, which makes it easy to assemble. The primary components of the housing are the inlet, outlet, carbon fiber tube adapter, housing-sensor holder coupler, sensor adapter, L-bend, and carbon-fiber tube.
**Inlet/Outlet:** The Inlet/Outlet of the sensor housing is designed to have a high cross-sectional area at the end of the inlet/outlet to maximize the airflow volume rate for a given pressure difference (low pressure on top of the propeller vs atmospheric pressure outside the UAS body). The reducer-expander strategy (funnel shape) allows the low-speed flow at the intake to flow at a higher speed proportional to the reduction of the cross-sectional area according to the continuity equation, VolumeFlowRate=Velocity∗Area=constant. The curved-horn shape of the part creates a smooth transition from a high cross-sectional area at the intake to the narrow cross section of the carbon-fiber tube which ensures laminar flow throughout the housing. The radius, *r*, of this curvature and gradient of this curve are chosen to minimize the loss of airflow and to maximize the flow velocity inside the tube. For the housing used in the paper inlet/outlet transitions from a diameter of 104 mm to 26 mm to match the carbon fiber tube, this reduction in diameter corresponds to a 16× increase in the speed from the inlet through the carbon-fiber tube.**Carbon-Fiber Tube Adapter:** This adapter adds on to the standard carbon-fiber tube to make them compatible with the mechanical screw-in threads of the other parts.**L-bend:** L-bends transition the flow in a perpendicular direction, allowing the safe extension of the sensor housing outwards parallel to the UAS arm while still using the propeller for air drawing. The curvature of the L-bend is minimized to reduce the flow loss due to the change of direction.**Sensor Holder Adapter:** This adapter is molded to fit on the sensor of interest. Due to mechanical screw-in threads, any sensor with an appropriate sensor adapter will be compatible with the housing. The sensor adapter, as shown in Figure 4b, is designed for iMet XQ2. Most of the sensors have strict storage requirements due to moisture affecting the quality of the sensor. Since the adapter is very easy to attach to the main housing, it can be plugged-in right before the UAS mission, keeping the sensor safe in storage otherwise. This *adapter*-type design choice allows several different sensor designs to be compatible without modifying the rest of the housing.**Housing-Sensor Holder Coupler:** This part provides an attachment point for the sensor adapter to attach into the housing. A sensor plugged into this adapter will be inside the tube aspirated by laminar airflow. Due to the modular design, multiple of these part can be connected in series to have an equal number of sensors taking a reading at the same time from the same sampling airflow.**Carbon-fiber tube:** The carbon-fiber tube diameter is critically the most important dimension, as it is the primary determinant of how much airflow is achievable through the tube and consequently drives the dimension of all the other parts. The Hagen–Poiseuille equation [37] dictates that the volume flow rate through a tube is quadratically proportional to its radius which means that for the same pressure difference and length of the tube, reducing the radius by half will reduce the volume flow rate to 1/16th the original. The diameter of the tube also needs to be big enough to allow the sensors to be in the center of the tube with sufficient area around it to promote good airflow. The diameter of the carbon fiber tube used in the paper is 26 mm.

**Modularity:** All the parts are designed to be modular for ease of replacement and manufacture. The modular design of the housing allows for the addition of multiple sensors to the housing at the same time, while the sensor adapter allows for the replacement of any sensor.

**Ease of production:** Since the parts are designed to be modular, it is also easy to be 3-D printed. All the parts except the carbon-fiber tubes are designed to be 3-D printed to reduce the cost of manufacturing and to make them readily reproducible and replaceable in the lab. Three-dimensional printing poses additional constraints on the design; for example, Fused Deposition Modeling (FDM) 3-D printers do not provide the same level of strength in the vertical axis of the part and usually have difficulty in printing slanted overhangs beyond 45${}^{\circ }$ angle. All the pieces are meticulously designed to satisfy these requirements, resulting in strong, lightweight, and reliable parts.

### 2.5. Experimental Methods

We designed our experiments to verify that the housing meets the design requirements (Section 2.1) and is capable of producing accurate and consistent measurements. The experimental measurements also evaluate the effect of different horizontal and vertical speeds as well as the orientation of the UAS.
EXP 1**Airflow Inside the Housing:** During a field test, the airflow speed inside the housing is measured using a hot-wire anemometer (Extech SDL 350, FLIR Commercial Systems Inc., Nashua, NH, USA) to establish the consistency of the flow speed and its directional dependence on ambient wind. We conduct ten vertical profiles at the same location within 13 minutes with successive 36${}^{\circ }$ changes in the orientation of the UAS between each ascent or descent. An analysis of this data would establish that the housing provides sufficient aspirations at all times, irrespective of the UAS altitude and relative orientation of the housing with atmospheric wind.EXP 2**MURC Tower Calibration:** Through the LAPSE-RATE field campaign, we compare our data against the ground-truth data obtained from the MURC tower, as shown in Figure 5. We perform these experiments to investigate (1) how accurately a sensor inside the housing captures the measurements in-flight when compared with the ground-truth and (2) whether the sensor is affected by external heat/radiation, resulting in measurement drifts.We use a similar flight pattern for both UASs (M600P1 and M600P2) by having the UAS climb up to the MURC tower altitude of 15.2 m and hover for 10 min. We also assume that the atmospheric parameters do not change within the horizontal distance of MURC and the UAS as shown in Figure 5b.EXP 3**Comparison of Flights at Different Vertical Speeds:** Due to the dynamic response characteristic and sensor response time, measurements made during ascent and descent at the same AGL altitude are expected to have a difference. This difference is proportional to the sensor response time and aspiration airspeed [20]. Since the housing is expected to maintain sufficient airspeed over the sensors at all times, the impact of the UAS vertical speed on the sensor response time should be negligible. Some recirculation of air from underneath the propeller is expected during hover or slow ascent/descent speeds. If the housing samples from the recirculated air, it will result in a measurement error or in a reduction of feature details in the observation. As such, we hypothesize that the housing would allow missions with faster ascending speeds, reducing the flight duration for similar altitude profiles. We test this hypothesis over ten flights with different ascent and descent speeds and compare the absolute error between the ascent and descent measurements. The UAS used in our experiments is limited to a maximum vertical speed of 5 m/s during ascent and 3 m/s during descent.EXP 4**Inversion Layer Tests:** We conduct the inversion layer tests to investigate whether the sensor is exposed to mixed and turbulent airflow of the UAS. Well-mixed air will not exhibit an inversion layer. If the UAS fly through an inversion layer and the TH sensor samples from the UAS mixed air, it will not be able to detect it. Hence, if a sensor inside the housing can detect an inversion at the true altitude, it will validate that the housing is not sampling from mixed air. Additionally, if there is an issue with aspiration, this will also make it harder to identify inversion. We perform the flights through an inversion layer (ground truth established using balloon radiosondes) at different ascent and descent speeds. We also conduct horizontal transects (at an altitude of 50 m) to investigate if the sensor readings are consistent when the sensor is in the wake of the UAS.
**Vertical Profiles:** We perform six flights at different ascent and descent speeds. We place the sensors in upstream (the sensor is mounted on the end directly facing the wind) of the UAS M600P1 and downstream (the sensor is mounted on the end facing away from the wind) of the UAS M600P2. Identification of the inversion layer with these profiles establishes whether the housing is sampling from the UAS’s disturbed airflow, the effect of the orientation of the UAS relative to the wind, and the presence of aspiration issues. Flights are also performed at different speeds to establish the effect on the sensor measurements.**Horizontal Profiles:** These experiments investigate the effect of ambient wind and the UAS’s relative orientation on the sensor measurements, especially for horizontal profiles. Our experiments were conducted such that the sensor is exposed either upstream (sensor on the leading front of the UAS) or downstream (sensor mounted on the trailing end of the UAS motion).**Comparison with Radio-Sonde Balloon:** We use two radiosonde balloons launched one hour apart as our source of ground truth for the inversion profiles. The release of the first balloon coincides with the first vertical profile, while the second balloon is launched at the start of the horizontal profiles. We use the data from the radiosonde for a comparison against the ascend data recorded by the UAS to assess how well it captures an inversion.

## 3. Results

The results are presented in subsections similar to that in the *Experimental Methods* (Section 2.5). We note that, for all the following analysis, the data from MURC and radiosonde are filtered by the manufacturers’ proprietary algorithms. However, the data from the UAS is not filtered or corrected for sensor response lag. The paper focuses on capturing the impact of the sensor housing on the observations/data collection. Filtering could wash out the difference in post-processing depending upon the details available in the observation. If the raw data does not capture the details accurately, post-process filtering would be impacted as well.

Assuming the ascent and descent profiles are identical at the same location within a reasonably short time-frame, they would appear symmetric/mirrored around the corrected profile if the sensor data is not corrected for sensor response lag. We do not perform such corrections for the data presented in this paper; standard post-processing can take them into account [30].

### 3.1. (EXP 1) Airflow inside the Housing

Figure 6 shows the actual airflow speed inside the housing during flights, as measured by an anemometer. The box plots show the distribution of flow speed for ten flights with different relative orientations of atmospheric wind over altitudes ranging from 5 m to 120 m above ground. As seen from the plots, the mean flow speed inside the housing is 12.45 m/s (with a variance of 1.25 m/s), which is roughly 2.5× the required aspiration for the iMet sensors used in the paper.

### 3.2. (EXP 2) MURC Tower Calibration

Figure 7 shows a time-series of the temperature (T) and humidity (RH) sensor readings for a 10 min hover along with the MURC tower readings. Since the sampling rate and exact time of data recording are not the same for all the sensors and MURC, all of the measurements from the UAS have been resampled linearly according to the time-stamp of the MURC data (with the assumption that the MURC and UAS clocks have no relative offset among them).

The time series shows that sensors used on the UAS are more sensitive to changes than those on the MURC and that the resolution of the temperature sensors is 10× of the MURC. A visual inspection also reveals that the sensors show a similar trend with MURC but with an artificial response time offset (MURC lags ∼40 s for the temperature sensors and ∼9 s for the humidity sensors).

If the time stamp on all the devices are accurate and synced using GPS, this could be a result of spatial variation where the horizontal wind carried a higher temperature air from a different place through these devices at different times. Alternatively, more likely due to the ill-matched time offset for temperature and humidity, the sensor response time of MURC could be higher, causing it to lag, especially if it is aspirated by atmospheric wind with an insufficient speed which may increase the response time. Another reason for the time offset could be the fact that a smoothing algorithm is in place for MURC with large moving average window. Another important thing to note is the humidity sensor response of P2Nimbus1 and P2Nimbus2. These two sensors, if plotted separately, show a similar trend as that of other sensors, but the sensitivity of the sensor appears to be very low and hence appears as a straight line when plotted together with the others. We see this effect throughout all the experiments where the humidity measurements of nimbus-pth are usually less sensitive to changes and could be at an offset with standard sensors. The data gathered from these sensors would still provide valid insights, and the effect could be corrected by tuning through the analog-to-digital conversion process for these sensors.

The absolute errors are within the bound of sensor uncertainty, as shown in the box plot in Figure 8, for sensors in the housing, whereas the sensor without housing (P2Nimbus1) has a higher range of errors with an increased variation in reading. The primary source of error in both figures is the difference in resolution and the artificial time offset, as discussed above. The key statistics of the measurement errors are presented in Table 3.

### 3.3. (EXP 3) Comparison of Flights at Different Vertical Speeds

This subsection presents the results from different ascent/descent speed combinations to find the ascent/descent speed pair that works best for the housing. Since we do not capture all ten profiles simultaneously and the variability in speed causes unequal data points among profiles, we plot TH against the altitude instead of plotting them as a time-series as we did for the MURC comparison. Figure 9 shows the temperature and humidity plotted against the altitude for five different combinations of ascent and descent speeds. Each row of the plot represents a different sensor as identified on the y-axis label, while the legend on the bottom of each column specifies the order in which we performed the flights. The title of each column of the plot represents the ascent and descent speeds. This figure also provides the grounds for comparison of the housed sensor with a non-housed sensor since P1XQ2 and P1Nimbus2 were inside the housing while P1Nimbus1 was mounted under the body of the UAS without housing. Although we performed the actual flight between 0–150 m AGL, both the ascent and descent data were trimmed to be 10–148 m and resampled at each meter of altitude to facilitate a comparison of the ascent vs. descent. The reason for trimming is to avoid a comparison of noisy data at the ground level and also to reduce the artificial inversion effect at the start and end of the flight due to the sensor’s dynamic response [20]. In Figure 9b, the humidity sensor on P1Nimbus1 and P2Nimbus2 seems to have an offset of ∼20% and, similar to the comparison with MURC, appears to be less sensitive to changes in the atmosphere when compared with P1XQ2. Despite the offset and sensitivity, the sensors still provide relevant features for a comparison based on the vertical speed. Due to the dynamic sensor response characteristic and corresponding time constant, a finite amount of difference is expected when we subtract the temperature profiles of the descent from the ascent (ΔT). This difference is expected to increase with the speed of ascent/descent and to decrease with the aspiration airspeed [20].

As seen in Figure 10a, sensors inside the housing are not affected by the vertical speed as much as the sensor without it. The ΔT on the housed sensor, which is sufficiently aspirated, increases significantly slower compared to the non-housed sensor as vertical speed increases. This increase in error is merely an artifact of the sensor response time constant, which prevents the sensor from *catching up* to larger step changes in temperature. The increase in the ascent/descent speed appears to add additional burden on the “effective” sensor response time, which would cap the maximum vertical velocity of the UAS for an accurate profile. However, the sensor inside the housing can exploit the maximum UAS capability for speed without significantly changing the response time.

Another interesting feature to note is that, when the descent is slower than the ascent, the difference appears to be lower than that of same ascent/descent speed (since a lower speed will produce values closer to the actual value). There is also an interesting trend in the humidity box plots in Figure 10b for P1XQ2: The range of absolute error decreases (while the mean error is still under 2%) with the increase in vertical speed. This is due to the fact that at, faster speeds, sensors do not pick up the fine-scale changes in atmosphere, making the data appear smoothed out. A comparison of the sensors shows an increase in the humidity error (ascent-descent) when the sensor is non-housed, which exacerbate as the vertical speed increases. It should be noted that the humidity sensor response time is dependent on the temperature, as specified in Table 2.

### 3.4. (EXP 4) Inversion Layer Tests

Figure 11 represents vertical inversion layer flights at different speeds of ascent and descent as indicated in the corresponding plots. The figure also features radiosonde data for the same altitude range that was launched a minute before the first UAS flight and 36 min after the sixth UAS flight. For this experiment, we consider only the sensors inside the housing. We note that the relative humidity data for P1XQ2 had to be discarded due to a glitch causing the sensor to be saturated at 100% before the inversion layer tests and hence is replaced by P1XQ1 relative humidity observations. The relative humidity for radiosonde is calculated from the dew point temperature ($T_{d}$) and air temperature (*T*) using the linear estimation formula as described in [38] to be $RH\approx 100-5*(T-T_{d})$.

The temperature profiles in Figure 11a demonstrate the ability to detect the inversion at similar altitudes as that of the radiosonde balloon deployed by NOAA. This confirms that sensor housing is not sampling from a disturbed air flow, since it would fail to detect the inversion otherwise. The performance of the sensor upstream (P1XQ2) seems to be better than that of downstream (P2XQ2) as the latter seems slightly warmer, which increases as the profile speed increases. These deviations could mean that, even though air is sourced far from the body, a sensor downstream from the wind could be exposed to waste heat from the UAS that is carried by the atmospheric wind (as discussed in Section 2.2). The effect is more significant during ascent if the ascent speed is significantly higher than the wind speed as the resultant wind vector points more towards the air source of the sensor located downstream. Consequently, in such cases, the descent profile matches more closely with the inversion profile.

The humidity profiles in Figure 11b match closely with that of radiosonde at slower vertical speeds. It should be noted that the operating temperature for these missions was low, which decreases the response time of the P2XQ2 humidity sensor. The slow sensor response time combined with the high vertical speed smooths out the data, and the humidity sensor loses the ability to identify a fine-scale inversion.

We performed four consecutive horizontal transects at three different speeds. The direction of each of the four transects along with the direction of air flow inside the housing and surface wind is shown in Figure 12. The sensor, P1XQ2, is downstream for transects 1 and 3 and upstream for transects 2 and 4 for UAS 1 (M600P1). The opposite configuration is used for P1XQ1. The sensor on UAS 2 is at a right angle with the direction of movement and hence not subjected to upstream or downstream UAS flow.

Figure 13 shows the time series for transects at three different horizontal speeds. For the temperature plots, a sensor that is not subjected to upstream or downstream flow (dashed line, P2XQ2) holds a fairly constant value over both directions of the transect. On the other hand, the sensor subjected to the downstream of UAS (solid line, transects 1 and 3) appears a bit warmer than the sensor placed upstream (solid line, transects 2 and 4). This finding agrees with that of the vertical profiles for the downstream sensors. The effect of sensor warming is more obvious at higher horizontal speeds. The statistics presented in Table 4 show an increase in the standard deviation for M600P1 in agreement with the above discussion.

For the relative humidity plots, surprisingly, the sensor subjected to upstream/downstream flow appears more consistent than the sensor not subjected to either stream. Although it is not certain as to what causes this deviation, all the fluctuations in TH readings are still within the sensor uncertainty.

## 4. Discussion

### 4.1. Housing vs. No Housing

A sensor without housing on a UAS has an increased *effective* sensor response time, as seen in the Figure 10, which increases with the vertical speed of the UAS. A sensor with a higher effective response time is incapable of resolving fine-scale changes and causes an undesirable “smoothing” of the data (Figure 9a). According to Figure 8, even for stationary purposes such as hovering in one place, a sensor without housing performs worse than a sensor inside the housing.

A sensor housing such as the one presented in this paper removes a lot of the uncertainty, as it always produces sufficient aspiration (Figure 6), is capable of operating at higher vertical and horizontal speeds (Figure 10 and Figure 13), and can detect inversion with a comparable performance as a radiosonde balloon (Figure 11).

Another significant advantage of the housing is its ability to reduce uncertainty in the descent data. Sensors located inside the housing demonstrate similar descent and ascent profiles in both well-mixed air (Figure 9) and through an inversion layer (Figure 11). These conclusions for the temperature and humidity sensors hold true even when comparing sensors that have similar specifications (such as P1Nimbus1 and P1Nimbus2). The observable difference between the ascent/descent for sensors inside the housing seen in all the plots is a result of dynamic sensor response characteristics and could potentially be corrected if the sensor characteristics are known. To the best of the authors’ knowledge, to date, no other housing for multirotors has demonstrated the capability of measuring useful data in both ascent and descent.

### 4.2. Effect of UAS Orientation on the Housing

***UAS Orientation Relative to Atmospheric Wind:*** The speed of the aspirating airflow inside the housing varies very little with the orientation of the UAS (Figure 6). At a lower speed of operation, a sensor housing downstream or upstream measures values unaffected by the UAS, while slight deviations could be seen when the sensor is located downstream from the UAS at a higher horizontal speed of operation (Figure 13). A similar effect could be observed when the sensor is downstream and the ascent speed of UAS is significantly higher than the horizontal wind speed: The ascent appears slightly warmer, which could be contributed by the waste heat of the UAS (Figure 11). It should be noted that. in both cases, the increase is not significant and is within the uncertainty bounds of the sensors. However, further experiments would be required to establish the effect of the waste heat on the measurements. This could also indicate that the inlet of the sensor housing should be placed even higher to avoid the warmer flow. Alternatively, a wind sensor can be used to reorient the UAS when the desired speed of operations is higher.

***UAS Orientation Relative to Sun:*** The primary difference between cloudy and sunny conditions will be the incident angle and intensity of solar radiation in the housing. Sensor probes are placed inside the housing, which protects it from incident solar radiation. The housing is made with very thin carbon-fiber material with a very low thermal mass; additionally, all parts are painted with reflective white paint, and solar reflective tape has been used on the carbon fiber tube. With the high aspirating air flow, highly reflective surface, and low thermal mass, the sensor housing is capable of reliable measurements in both cloudy and sunny conditions in any UAS orientation relative to the sun.

### 4.3. Effect of Housing on the UAS

The UAS used in this paper is capable of a maximum 5 m/s in ascent, 3 m/s in descent, and 10 m/s in the horizontal direction when used in automatic missions. A stricter limit on the descent is due to the fact that, as the UAS descends into the turbulent downwash, it may start oscillating uncontrollably [9], resulting in a loss of control or crash. Due to the nature of the housing presented here, it extends out 0.7 m (2.6 times the propeller radius) away from the center of rotor. This requires a second housing to be mounted on the opposite side as well to balance the center of gravity. This comes at a cost of decreased sensitivity of the UAS in the roll/pitch axis (mounting axis). Traditionally, this sensitivity could be improved by tuning the flight control parameters. However, that approach is not explored in this paper to keep the sensor housing generic to commercially available UASs.

Although the structure is quite large, it is very lightweight. There was no loss of control during the 21.7 h of flight time recorded in 170 flights by two vehicles during the LAPSE-RATE campaign. However, for some of the flights, when the vehicle went through turbulent regions of the atmosphere, there was noticeable wobbling in the roll axis (the mounting axis), lasting up to 30 s. Although rare (8 out of 170 flights), the effect would appear more on the descent than on the ascent as the UAS descent on the turbulent air field created by its own propeller. An onset of this oscillation can be detected early on using an on-board computer, as described by Reference [39] and can be countered by slowing down the multirotor speed when passing through turbulent regions of atmosphere.

Considering these limitations of the UAS, even though the housing is proven to be capable of operating at higher speeds (as seen in the results and the preceding discussion), a near-optimal range of operation would be 0.5–3 m/s for the ascent, 0.5–2 m/s for the descent, and 0.5–5 m/s for the horizontal transect.

### 4.4. UAS Ranges of Operation

Typical ABL profiles are up to 2 km, and the Federal Aviation Administration (FAA, USA) certification of authorization for LAPSE-RATE was valid for up to a 1 km flight. However, software restrictions on the autopilot of UASs limited our profiles to 500 m AGL. The sensor housing is capable of reliable measurements at various temperature/altitude ranges. Data has been collected in different ranges of temperature in various weather conditions and times of day. To demonstrate the altitude and temperature range capability, additional data collected during the LAPSE-RATE campaign is presented in Figure 14. The first subplot in Figure 14 shows a temperature profile up to 500 m with a temperature difference of ∼5 degree Celsius between 10–500 m altitudes at noon when the atmosphere is well-mixed; the second subplot shows inversion detection for a flight up to 300 m before sunrise. Also, the data presented in the Results section such as Figure 9 and Figure 11 has a 15 degree Celsius difference in the operating temperature.

### 4.5. Limitations and Future Work

***Sensor Housing Future Improvements:*** The sensor housing as proposed shows the ability to maintain a sufficiently high aspiration at all times. However, some variability in the speed of aspiration still exists as a function of varying rotor speeds (and possibly ambient wind). If a sensor’s response time is highly sensitive to the variation in airspeed, active control of the airspeed might be required. Alternatively, design changes could be made to add passive gills that will resist sudden fluctuations in speed inside the housing.

Descent profiles using traditional methods are unreliable compared to an ascent profile due to the turbulent mixing of air by the UAS while descending. While our sensor housing is able to obtain reliable descent data that is within the sensors’ uncertainty, some features in the data are still not fully captured, as evident from Figure 9. This would indicate that some turbulently mixed air is still passing through the housing inlet. Extending the housing even farther out would reduce the effect of mixed air further and would reveal more features in the observations. We note that any extension of the sensor housing would burden the flight controller as well and should be done according to the UAS’s safety limit.

***Sensor Redundancy:*** The sensor housing shields the sensor from external heat sources, but when located downstream from a heat source (such as the waste heat of the UAS), the effect is usually unavoidable. For the cases encountered in our experiments, the effect is not significant. If the UAS is equipped with sensors in the opposite arm, this could be solved in two ways: (1) a reorientation of the UAS (using data from an onboard anemometer) or (2) sensor fusion using the redundant data based on the wind/UAS speed and direction.

***UAS Safety Maneuvers:*** The detection of turbulence and oscillations based on real-time data could allow the UAS to perform preprogrammed safe maneuvers to avoid a loss of control. This would allow for the safer operation of UASs at higher speeds, making the sampling process significantly faster.

***Correction of Measured Data:*** The correction of sensor data can be complex without accurately characterizing the sensors. The sensor response time of the iMet XQ, for example, is based on the step response of the sensor while subjected to an oil bath. The step response information alone is not enough to correct for the dynamic response of the system. A robust method to characterize the dynamic response of the UAS/housing/sensor system would be needed to account for the deviation or to make any corrections since the system response time would be different for just the sensor vs. a sensor inside the housing *as flown* [30].

Another challenge in correcting the sensor data (such as the humidity) is the variability in response time based on temperature which should be taken into account for the correction routine. Additionally, as noted in previous literature, multirotors have a dispersion effect, and recirculating air can affect the measurement by changing the *actual* observation level. We suspect this will be more apparent in lower ascent/descent speeds as the air has a higher chance of being recirculated. These effects can be mitigated further by extending the air inlet even farther out from the body.

## 5. Conclusions

In this paper, we have presented the design and evaluation of a novel sensor housing that allows multirotors to reliably host atmospheric sensors over various profiling missions. The design passively induces a high-speed laminar airflow inside the housing by using the pressure deficits created by the propellers, and samples air from a source far removed from the strongly-mixed turbulent flow. We evaluated the nature of the airflow and empirically showed the effectiveness of the housing in maintaining a strong airflow (≥ 10 m/s) invariant to ambient wind. Numerous field tests conducted at the LAPSE-RATE’2018 flight week demonstrated the ability of the housing design to precisely capture variations in atmospheric observations, while providing reliable sensor performance during high-speed ascent *and* descent.

## Figures and Tables

**Figure 1 sensors-19-02481-f001:**
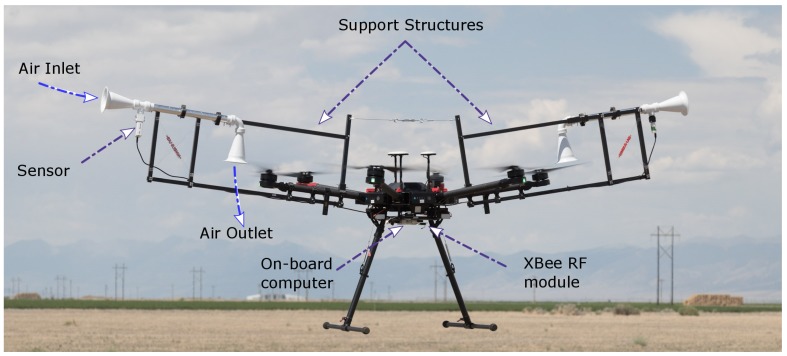
A field trial of sensor housings mounted on a *DJI M600 Pro* UAS using the square carbon-fiber tube support structure.

**Figure 2 sensors-19-02481-f002:**
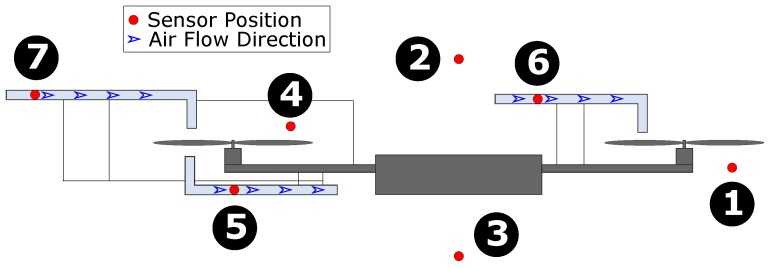
A schematic of the different Sensor Placement Configurations: ❶ Direct Downwash, ❷ Over the UAS, ❸ Under the UAS, ❹ Direct Upwash, ❺ Downwash housing, ❻ Upwash Housing with Inlet Pointed Inside, and ❼ Upwash Housing with Inlet Pointed Outside.

**Figure 3 sensors-19-02481-f003:**
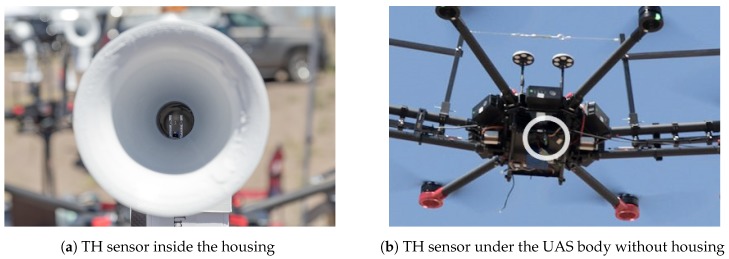
The sensor configurations used in the experiments for this paper: The sensor without housing is mounted without any additional radiation shielding.

**Figure 4 sensors-19-02481-f004:**
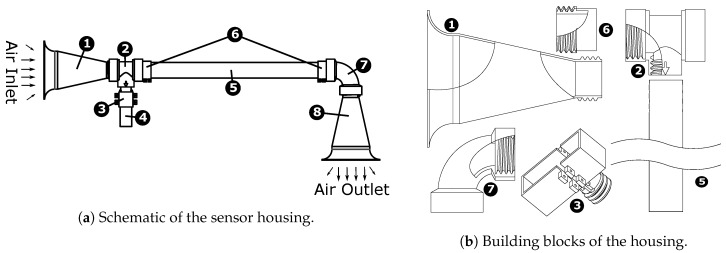
❶ Inlet, ❷ Housing-Sensor Holder Coupler, ❸ iMet XQ2 Sensor Adapter, ❹ iMet XQ2 sensor, ❺ Carbon Fiber Tube, ❻ Carbon Fiber Tube Adapter, ❼ L-bend, and ❽ Outlet.

**Figure 5 sensors-19-02481-f005:**
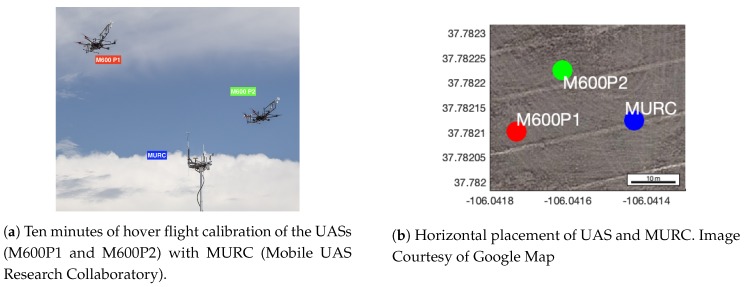
MURC tower calibration.

**Figure 6 sensors-19-02481-f006:**
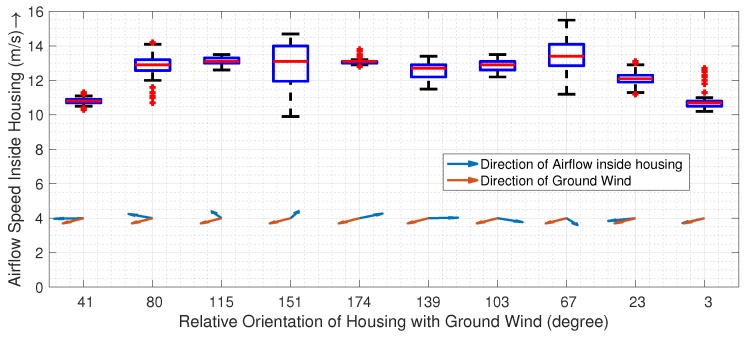
The plot shows a sensitivity in the airflow speed inside the housing in the presence of wind for 10 consecutive flights at different orientations of the UAS from the north at a ground wind speed of 6 mph from the northeast. The box plot captures the 25th to 75th percentile of data, while the central red line indicates the median. The black whiskers and the red plus signs indicate the extreme range and outliers respectively.

**Figure 7 sensors-19-02481-f007:**
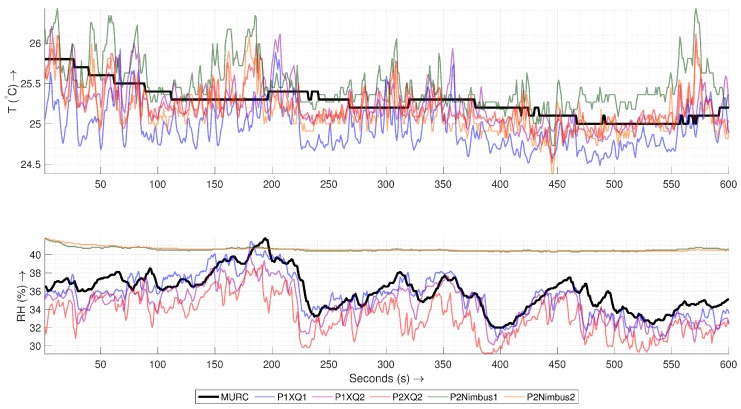
The Temperature (T) and Humidity (RH) time series of the MURC tower and other sensor readings for the 10 min of hover of the UAS at the MURC tower altitude (15.2 m): Only the data from MURC is filtered and corrected for sensor response.

**Figure 8 sensors-19-02481-f008:**
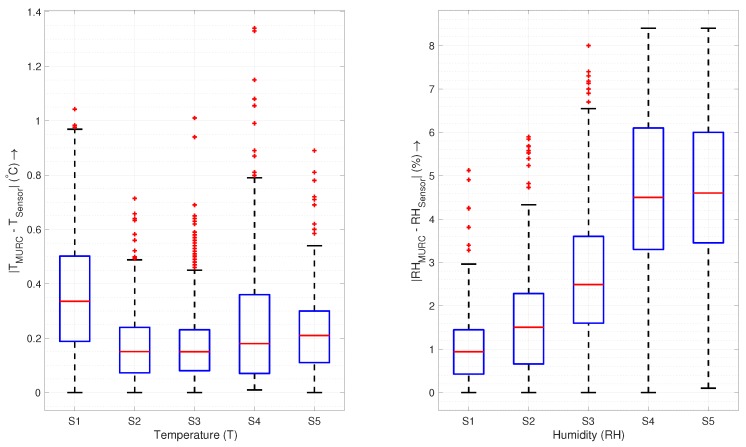
A comparison of the sensor readings with the MURC tower: The data presented here summarizes 10 minutes of hover at the MURC tower altitude (15.2 m) as presented in Figure 7. (S1—P1XQ1 (housing), S2—P1XQ2 (housing), S3—P2XQ2 (housing), S4—P2Nimbus1 (no housing), and S5—P2Nimbus2 (housing)).

**Figure 9 sensors-19-02481-f009:**
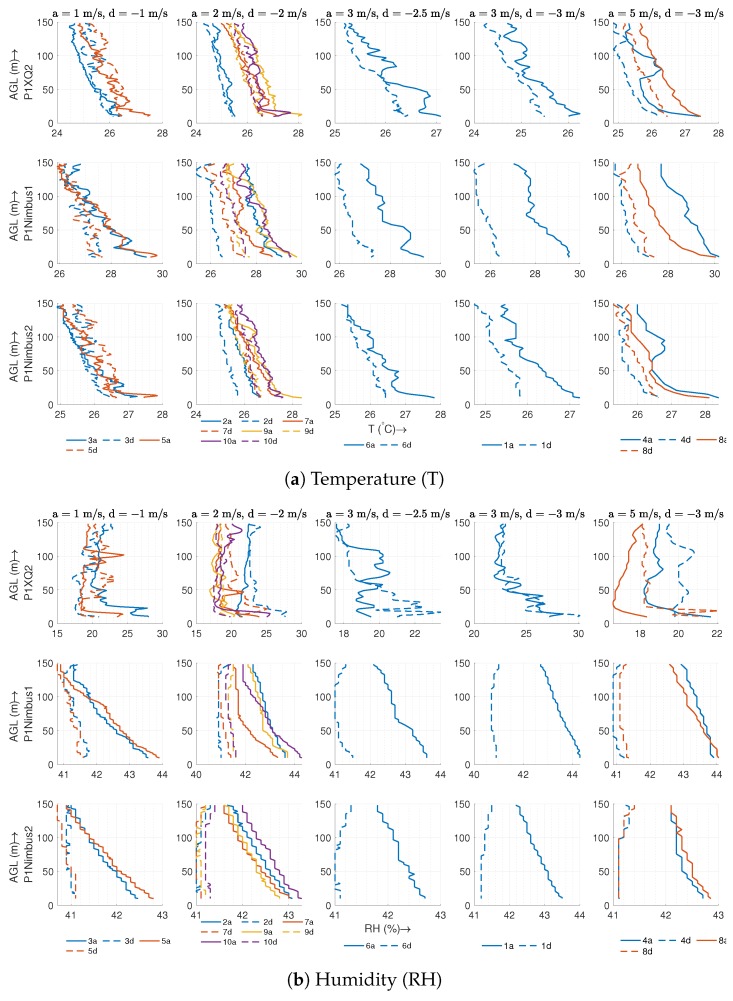
The plot shows the ascent (**a**) and descent (**b**) temperatures against the altitude for 10 consecutive flights conducted over a period of 60 min (7/18/2018 7:21 pm UTC–8:20 pm UTC) at different *“a”* and *“d”* speeds mentioned in the title of each subplot. Multiple flights at the same speed are grouped together. The colored legend under the plot indicates the order in which the flight was performed. The sky was slightly cloudy with cumulonimbus clouds, and the ground wind was 4–5 m/s from the NW. The data presented in this plot is not filtered or corrected for sensor response characteristics.

**Figure 10 sensors-19-02481-f010:**
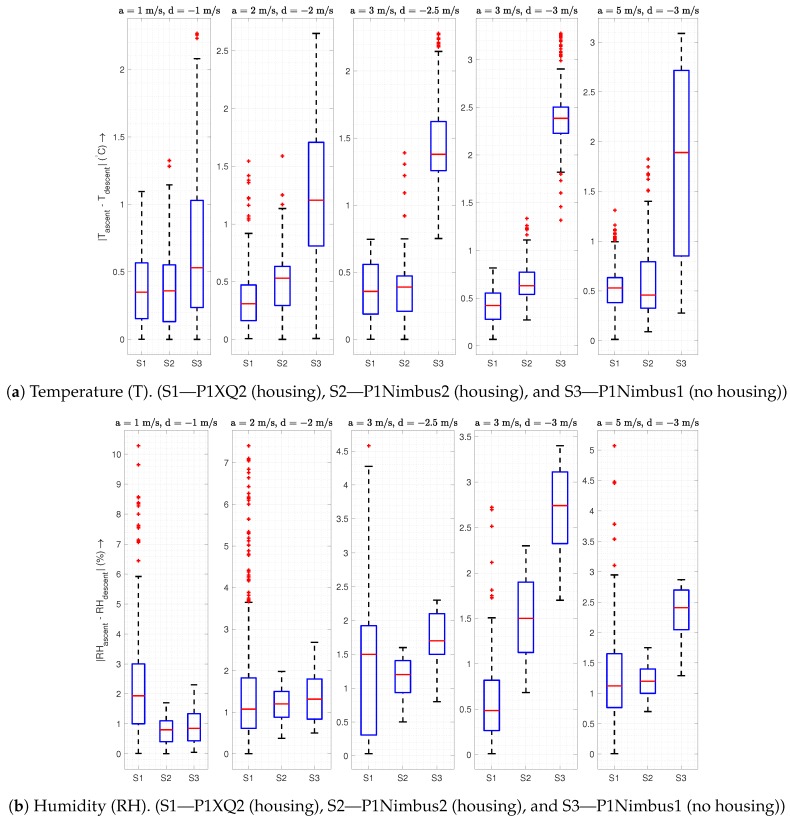
The box plot summarizes Figure 9 by showing the absolute (ascent − descent) temperature and humidity differences for 10 consecutive flights conducted over a period of 60 min (7/18/2018 7:21 pm UTC–8:20 pm UTC) at different ascent (**a**) and descent (**b**) speeds mentioned in the title of each subplot. Multiple flights at the same speed are grouped and plotted together. The sky was mostly sunny with cumulonimbus clouds, and the ground wind was 4–5 m/s from the NW.

**Figure 11 sensors-19-02481-f011:**
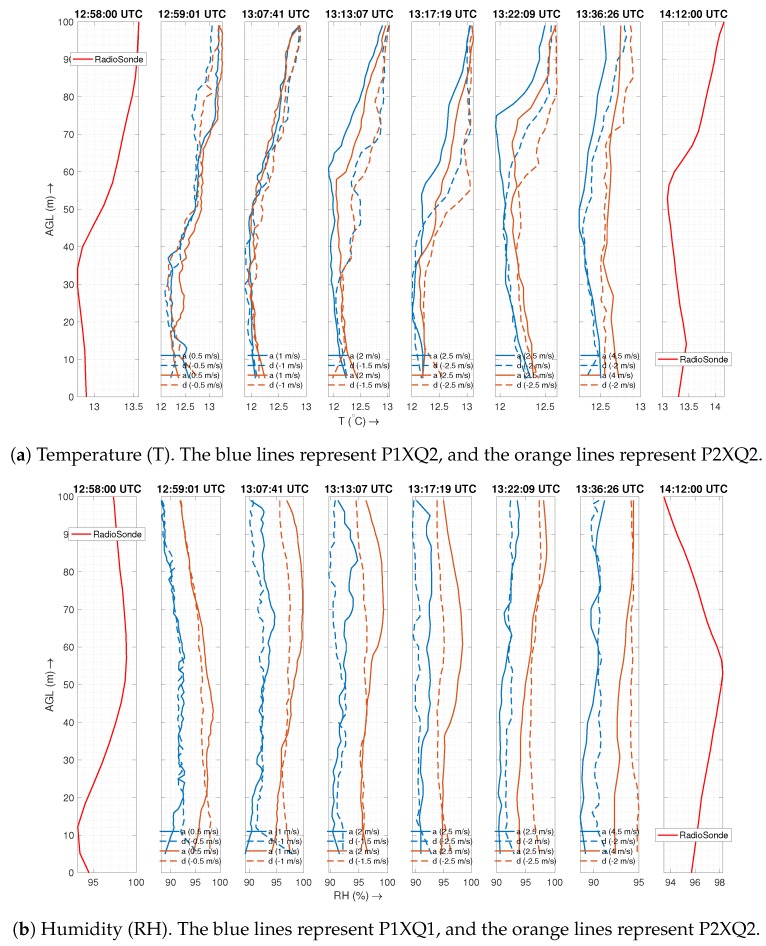
The plot shows two radiosonde flights (filtered and corrected) and six different flights of the UAS (not filtered/corrected) at different speeds in a inversion layer. The surface wind was 2 m/s from 290–308${}^{\circ }$ relative to north and with a partly cloudy sky. The P1XQ2 sensor is upstream, while the P1XQ1 and P2XQ2 sensors are downstream from the UAS with respect to wind.

**Figure 12 sensors-19-02481-f012:**
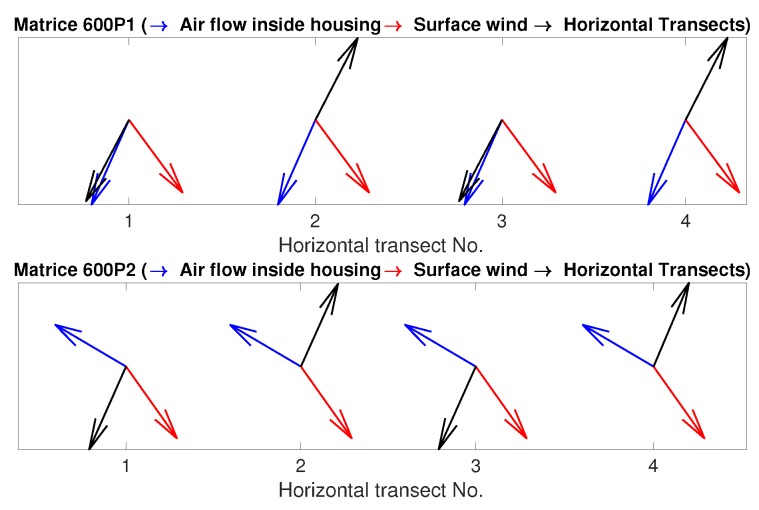
The figure shows the air flow, surface wind, and transect direction during each of the four transect flights conducted during each of the three horizontal flights through the inversion layer.

**Figure 13 sensors-19-02481-f013:**
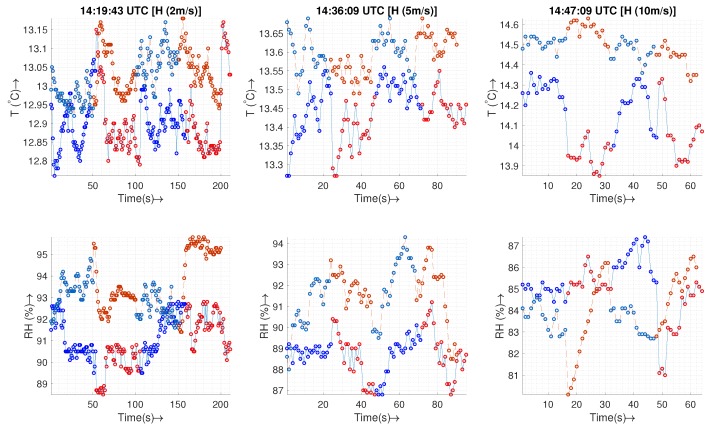
The plot shows three different horizontal “transect” flights at different horizontal (H) speeds at 50 m AGL. The surface wind was 2 m/s from 290–308${}^{\circ }$ relative to the north, and the sky is partly cloudy. The solid line plots indicate data from M600P1, and the dashed lines indicate data from the M600P2. The blue (M600 P1) and light blue (M600 P2) scatter dots indicate transects 1 and 3, while the red (M600 P1) and orange (M600 P2) are transects 2 and 4. Figure 12 shows the direction of air flow inside the housing, the surface wind, and the horizontal transects for each of the transects for both UASs. The data shown here is not filtered/corrected for sensor response characteristics.

**Figure 14 sensors-19-02481-f014:**
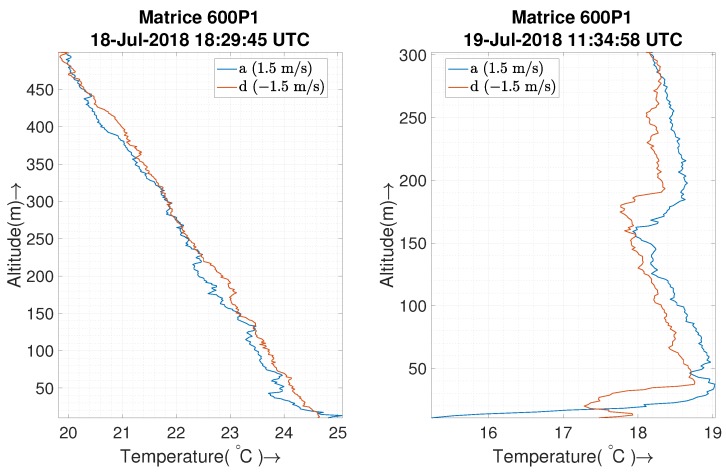
The first subplot shows unfiltered temperature data collected from 10 m–500 m of a well-mixed atmosphere at noon with a ground wind of 1.6 m/s with a 5 m/s gust.

**Table 1 sensors-19-02481-t001:** A comparison of the different sensor configurations: ✓ the availability of aspiration for temperature–humidity (TH) and atmospheric shielding.

No.	Configuration	Temperature	Humidity	Shielding	Comments
1	Direct Downwash	✓	✓	x	Mixed air; reading is especially unstable during descent
2	Over the UAS	x	x	x	Insufficient aspiration
3	Under the UAS	x	x	x	Mixed air and insufficient aspiration
4	Direct Upwash	✓	✓	x	Possibly insufficient aspiration
5	Downwash housing	✓	✓	✓	Mixed and turbulent Air
6	Upwash Housing with Inlet Pointed Inside	✓	✓	✓	Air source is in the wake of the UAS
7	Upwash Housing with Inlet Pointed Outside	✓	✓	✓	Air is sourced outside of the UAS interference

**Table 2 sensors-19-02481-t002:** The key specifications for the sensors used in different experiments: The unavailable fields are left blank.

		XQ2	XQ1	nimbus-pth	MURC	RadioSonde
		(iMet XQ2)	(iMet XQ1)	(Custom Built)	(Gill MetPak Pro)	(RS92)
Temperature	Type	Bead Thermistor	Bead Thermistor	Bead Thermistor	pt100 1/3 Class B	Capacitive Wire
	Range	−90 to 50 ${}^{\circ }$C	−95 to 50 ${}^{\circ }$C	−40 to 100 ${}^{\circ }$C	−50 to 100 ${}^{\circ }$C	−90 to 60 ${}^{\circ }$C
	Response Time	1s @ 5 m/s	2 s		<0.5 s	<0.4 s @ 6 m/s flow
	Resolution	0.01 ${}^{\circ }$C	0.01 ${}^{\circ }$C	0.01 ${}^{\circ }$C	0.1 ${}^{\circ }$C	0.01 ${}^{\circ }$C
	Accuracy	±0.3 ${}^{\circ }$C	±0.3 ${}^{\circ }$C		±0.1 ${}^{\circ }$C	0.5 ${}^{\circ }$C
Humidity	Type	Capacitive	Capacitive	Capacitive		Capacitive
	Range	0–100% RH	0–100% RH	0–100%	0–100% RH	0–100% RH
		@ 25 ${}^{\circ }$C, 0.6 s	5 s @ 1 m/s velocity	8 s		@ 20 ${}^{\circ }$C, <0.5 s
	Response Time	@ 5 ${}^{\circ }$C, 5.2 s				@ −40 ${}^{\circ }$C, <20 s
		@ −10 ${}^{\circ }$C, 10.9 s				@ 6 m/s
	Resolution	0.1% RH	0.7% RH	0.01% RH	0.1% RH	1% RH
	Accuracy	±5% RH	±5% RH	±2%	±0.8%	5% RH

**Table 3 sensors-19-02481-t003:** The key statistics for the MURC calibration error (MURC−Sensor).

Sensor		Temperature (${}^{\circ }$C)			Humidity (%)	
	Mean	Variance	RMS	Mean	Variance	RMS
P1XQ1	0.32	0.064	0.41	0.51	1.324	1.26
P1XQ2	0.05	0.043	0.21	1.51	1.546	1.96
P2XQ2	0.07	0.048	0.23	2.70	2.266	3.09
P2Nimbus1	−0.22	0.075	0.35	−4.56	3.915	4.97
P2Nimbus2	0.13	0.059	0.28	−4.58	3.870	4.99

**Table 4 sensors-19-02481-t004:** Statistics for the measurements presented in Figure 13.

UAS Platform	Horizontal Speed	Temperature (${}^{\circ }$C)		Humidity (%)	
		Mean	Std	Mean	Std
	2	12.90	0.08	91.03	1.09
M600 P1	5	13.44	0.06	88.71	0.98
	10	14.11	0.15	85.04	1.30
	2	13.02	0.06	93.46	1.20
M600 P2	5	13.59	0.05	91.59	1.41
	10	14.5	0.06	83.86	1.37

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
