# Peer review of "Design and Evaluation of Sensor Housing for Boundary Layer Profiling Using Multirotors"

_sensors, 2019, doi:10.3390/s19112481_

Round 1

Reviewer 1 Report

The results are relevant and I recommend this manuscript for publication in sensor after addressing some minor revisions.

-     Please add the outline of your paper at the end of your introduction. “The rest of this study is organized as follows: the materials and methods are presented in Section 2. In Section 3, …”

-      Please write the title of tables in the top of tables.

Author Response

Thank you for your review of our paper. Please see the attached pdf for the description of the changes that we made based on your feedback.

Reviewer 2 Report

Attached.

Author Response

(The authors gave the same response as above.)

Reviewer 3 Report

This paper is in general hard to read and gets lost in a large number of graphs and very lengthy explanations.

·         The base-line for comparison is balloons, but temperature and humidity can be measured accurately with lidar from the ground. This needs to be discussed and justified why a lidar is not used as the baseline. E.g. how does the tower compare to lidar measurements?

·         The filtering process on MURC is not explained very well. How accurate is MURC? Should apply similar filters on the UAS sensors so can more accurately compare.

·         Comparisons are made between with and without the housing, but it looks like unfiltered data is used for both. What gains are made with the housing when compared to a filtered non-housing signal? There is likely very little difference, which means that the only potential benefit is real-time measurement, but it’s not clear whether this would have any application when the data is post-processed anyway.

·         Should use Degrees or Fahrenheit instead of Kelvin.

·         There are only very small changes in temperature overall of around 299K which is close to room temperature. What happens with much lower or higher values? It’s impossible to evaluate the impact of the housing with such a narrow range of temperature tests.

·         What are the wind speeds? There needs to be tests with different wind speeds. Also what if it’s raining or cloudy versus sunny conditions?

·         The Atmospheric Boundary Layer is typically up to about 2 km altitude. These altitudes are likely hard to achieve due to airspace restrictions and need to be discussed.

·         How well does the method measure wind speed? This is probably the most important measurement. E.g. can it detect turbulence or shear layers in the atmosphere over time?

·         What extra weight does the sensor add?

·         It’s inadequate to say a comparison with the industry sensor is not within the scope. This was included on the UAS so should include in the results as it’s an important comparison to make.

·         What are the correlations for all the observations on Figure 13? It’s very hard to tell from the box plots alone.

·         Should show a time series comparison and state the R^2 values.

·         The Humidity results overall are not convincing. In a number of cases the housing is worse than non-housing, particularly for the lower a and d speeds.

·         Figure 16 is not relevant as it’s within the sensor uncertainty. Could just give mean and standard deviation.

Author Response

(The authors gave the same response as above.)

Reviewer 4 Report

·      L 14: „good agreement between ascent and descent profiles prove our unique design to be reliable for various UAS missions”: This is in contrast to the profiles in fig. 12 showing mostly non-matching profiles for ascent and descent.

·      L 49: “… consistent aspiration throughout the mission … “: During an ascent, the wind speed created by the multitor’s propellers should be different if compared to the descent or a horizontal flight and furthermore depend on the ambient wind speed. This should have an impact in the data and/or response times. How can you take this into account when analyzing the data? As a consequence, lines 53-54 seems to be very ambitious (“outperforms traditional configurations …. In terms of data consistency, correctness and reliability.”).

·      L 148: The manufacturer’s data provided for the GPS accuracy seems to be very optimistic, especially for the vertical accuracy. Has this been confirmed/tested by other means/instruments?

·      L 296: “Since well-mixed air will not exhibit an inversion, a sensor inside the housing that can detect inversion at the true altitude will validate the housing.” What is the true altitude? Is the balloon your reference? This might be a weak point, as the sensors of the balloon might have a slow response time as well. Only sensors with a fast response time (<< 1 s) will help decide where the “true altitude” of the inversion is located. 

·      L 302 ff: The design of the experiments is very good, but the deductions are not supported by the results.

·      Caption of fig 6: There seems to be no flow within the housing which should be the main goal of the CFD simulations. In 6 b) and c) it is not clear why the results are non-symmetric – is it due to computational errors?

·      Fig 7: Perhaps I do not understand this figure. If I catch it right, there is a dependency on the ambient wind direction within the housing – this would be quite bad as the sensor’s response time would vary on the ambient wind and the data analysis would be quite hard.

·      Fig 12: There is nearly no matching of ascent and descent profiles. The descent profiles show much less features which is probably a hint to the non-ideal ventilation/aspiration of the sensors. This is in contrast to the ambitions as described in the abstract.

·      L 414: This would mean that the motors (as the main source of heat) of the RPAS have a high rate of self-heating and this heat is transferred to the sensor within the very short time of being ventilated near the sensor. Is there any hint that this really happens or is it a hypothesis?

Author Response

(The authors gave the same response as above.)

Round 2

Reviewer 2 Report

Authors have made adequate changes per reviews.

Author Response

Thank you for your detailed review of our revised manuscript.

Reviewer 3 Report

A very extensive rebuttal and the changes have significantly improved the paper. It's good to have practical experiments and a very detailed analysis. It's acceptable in its current form.

Author Response

(The authors gave the same response as above.)

Reviewer 4 Report

Dear authors,

thanks a lot for the revision which has improved the paper noticeable. There are still some minor things to mention:

a) The text - especially the new part - should be re-checked for minor errors.

b) The captions of the relevant figures should reflect if the response time of the sensors have been taken into account (or have not been taken into account).

c) The heat transfer from motors or batteries is actually very much depending on the specific UAS configuration (including payload). Providing the reference to the work of the Oklahoma group is ok, but the transfer to other (including your) systems is debatable.

Author Response

Thank you for your detailed review of our revised manuscript. Here's the list of changes we made according to your suggestions:

a) The text - especially the new part - should be re-checked for minor errors.

We have re-checked and corrected for minor errors in the entire paper. 

b) The captions of the relevant figures should reflect if the response time of the sensors have been taken into account (or have not been taken into account).

We have added details of sensor response time correction (or lack of it) on the captions. 

c) The heat transfer from motors or batteries is actually very much depending on the specific UAS configuration (including payload). Providing the reference to the work of the Oklahoma group is ok, but the transfer to other (including your) systems is debatable.

We have clarified the discussion to reflect that the effect of waste heat on our housing is a hypothesis and further experiments are required to determine its validity.